# OBJECT-LEVEL DATA AUGMENTATION FOR VISUAL 3D OBJECT DETECTION IN AUTONOMOUS DRIVING

## ABSTRACT

Data augmentation plays an important role in visual-based 3D object detection. Existing detectors typically employ image/BEV-level data augmentation techniques, failing to utilize flexible object-level augmentations because of 2D-3D inconsistencies. This limitation hinders us from increasing the diversity of training data. To alleviate this issue, we propose an object-level data augmentation approach that incorporates scene reconstruction and neural scene rendering. Specifically, we reconstruct the scene and objects by extracting image features from sequences and aligning them with associated LiDAR point clouds. This approach is intended to conduct the editing process within a 3D space, allowing for flexible object manipulation. Additionally, we introduce a neural scene renderer to project the edited 3D scene onto a specified camera plane and render it onto a 2D image. Combining with the scene reconstruction, it overcomes the challenges stemming from 2D/3D inconsistencies, enabling the generation of object-level augmented images with corresponding labels for model training. To validate the proposed method, we apply our method to two popular multi-camera detectors: PETRv2 and BEVFormer, consistently boosting the performance. Codes will be public.

## 1 INTRODUCTION

Data augmentation has become a necessary part of successful applications of visual-based autonomous driving (AD) 3D object detection, as it benefits the data diversity and model generalization ability(Yang et al., 2022b). Recent visual-based AD detection works typically employ image manipulations (Cubuk et al., 2018) and BEV (Bird's Eye View) augmentation (Huang et al., 2022) in their workflows (as depicted in Fig. 1-a,b), but these methods are limited to the image or BEV level, insufficient for enhancing scene diversity to match the complexity of real-world driving scenarios. Therefore, object-level augmentation techniques are essential to enrich scene variety.

Unfortunately, it is non-trivial to apply existing object-level augmentation techniques (Dwibedi et al., 2017; Yang et al., 2022a; Zhang et al., 2020) to visual-based 3D detection. When operating objects in 3D space, such as translation and rotation, objects' new appearance on the image usually cannot be obtained by straightforward 2D transformations (as depicted in Fig 1-c). We need to deal with thorny issues including but not limited to occlusion calculation, object's novel view synthesis, background inpainting and multi-camera consistency.

Aiming to address the above problems, we formulate object-level augmentation in 3D space as a dual process of **reconstruction** and **synthesis** (as depicted in Fig. 1-d). This is inspired by the easy-to-edit nature of LiDAR point clouds (Fang et al., 2021; Reuse et al., 2021), which could serve as a bridge between 2D images and the 3D world via projection, allowing edits in the point cloud domain to impact 2D images. In the **reconstruction step**, we generate dense point clouds of background and objects. We then transfer information from the image sequences onto these point clouds through training, facilitating scene reconstruction using point clouds as a intermediary. Specifically, to enable the point cloud effectively encapsulate contextual scene information, each point is assigned with a learnable neural descriptor. Additionally, we utilize a trainable neural scene renderer to recover images from the dense point clouds and their associated descriptors. Supervised by 2D RGB images, trained neural descriptors and the scene renderer could support high-quality reconstruction and rendering of the entire scene. In the **synthesis step**, we conduct the flexible manipulation of the scene through object point clouds. By seamlessly integrating object point clouds

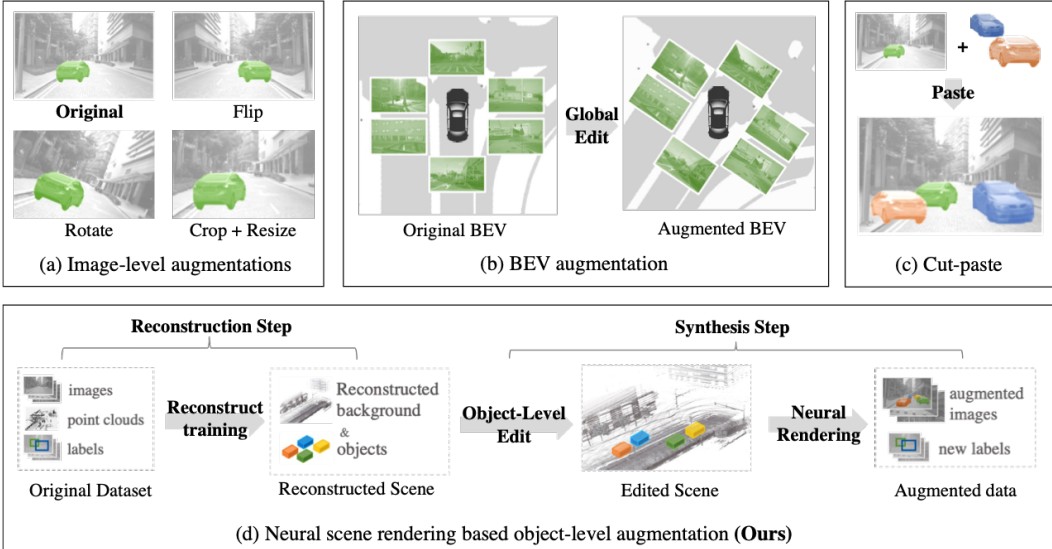

Figure 1: Comparison between different data augmentation methods for visual 3D detection. Image-level augmentations(a) primarily benefit the image backbone. BEV augmentation(b) transforms the extracted BEV layout but cannot create new layouts. Cut-paste(c) (Dwibedi et al., 2017) is limited to 2D manipulation and unsuitable for complex editing. Our approach (d) employs neural rendering for 3D scene reconstruction and object editing, ensuring 3D-2D consistency and producing higher-quality object-level augmented data.

into the scene point cloud, we achieve customization of scene content on a 3D level. Utilizing the descriptors and renderer trained in the reconstruction step, augmented images along with their corresponding labels can be obtained.

The proposed method can effectively address the aforementioned issues: By conducting object-level editing in 3D space and projecting it onto a 2D plane, the result images naturally adhere to principles of perspective and occlusion. Furthermore, overlapping information from different views and temporal sequences in the data set allows for detailed scene and object reconstruction from various angles, enabling object novel view synthesis and occluded background completion. For multi-view perception data, different cameras capturing the same scene data ensure object consistency within synthesized multi-view images.

In summary, our contributions can be listed as follows: **First**, we introduce an object-level augmentation framework for visual-based 3D detection, which employs a two-step process: reconstruction and synthesis. This framework is able to serve as a fundamental component for existing 3D detection methods, to increase the training data diversity. **Second**, we employ neural scene rendering to realize the aforementioned framework, offering an effective solution to mitigate the 2D-3D inconsistency issues. Following the training of the neural scene renderer and point cloud descriptor, we can seamlessly carry out object-level augmentation within the reconstructed scene. **Third**, we evaluate the proposed augmentation method on the widely-used nuScenes dataset(Caesar et al., 2020), demonstrating its capacity to enhance visual 3D object detection performance.

## 2 RELATED WORK

### 2.1 VISION-BASED 3D OBJECT DETECTION

Vision-based 3D object detection Zhang et al. (2021); Lu et al. (2021); Reading et al. (2021); Park et al. (2021); Ma et al. (2020); Peng et al. (2022); Xu et al. (2023) is a long-standing task in computer vision. The variant in multi-view setting has attracted much attention recently. BEVDet (Huang et al., 2022), BEVDepth (Li et al., 2023) are typical works of Lift-Splat-Shoot Philion & Fidler (2020)(LSS) paradigm, which utilizes a depth distribution to model depth uncertainty and project

multi-view features into the Bird's Eye View (BEV) space. DETR3D (Wang et al., 2022) and PETR (Liu et al., 2022a) perform end-to-end learning from the perspective of interaction between object queries and multi-view image features. BEVFormer (Li et al., 2022a) adopts explicit predefined grid-shaped BEV queries to generate BEV features for 3D detection.

## 2.2 STREET VIEW SYNTHESIS

Street view synthesis is a sub-variant of Novel View Synthesis Mildenhall et al. (2021), which tries to synthesize novel view from sequential image data of street scenes. Considering the rare-overlap characteristic in such data, several works Xie et al. (2023); Tancik et al. (2022) based on Neural Radiance Field (NeRF) (Mildenhall et al., 2021) have been proposed to deal with the rare overlap problem. Barron et al. (2021); Tancik et al. (2022); Rematas et al. (2021) are trained with specially collected dataset, which allows a large amount of overlap. Although these methods are able to reconstruct the street scenes and render novel street views, they still suffer from the dataset with sparse input views, and easy to synthesize images with artifacts and blurs. Xie et al. (2023) propose S-NeRF to reconstruct the street scenes with sparse Lidar points and depth error. READ Li et al. (2022b) introduce point-based neural descriptors to attach image features to point cloud, achieving relatively stable synthesis results in large-scale scene.

## 2.3 DATA AUGMENTATION FOR VISION-BASED PERCEPTION

Data augmentation is a technique commonly used in deep learning and computer vision. In vision-based perception area, data augmentation techniques are usually conducted in image level. Random scale, crop, flip operations are widely used to training images to increase the diversity of data Cubuk et al. (2018); Yang et al. (2022b). Huang et al. (2022) disturbances the generated BEV features and introduces BEV augmentation which significantly imporves the final performance of detector. Only a few works have explored object-level augmentation techniques for vision-based perception. Santhakumar et al. (2021) introduces 2D object-level method cut-paste for monocular object detection, they cut objects from the source frame and paste them to the target frame to generate new data.

Recently, several works focus on reconstructing and editing 3D scenes. UniSim Yang et al. (2023) and Mars Wu et al. (2023) build a neural sensor simulator with the input scene sequential data and conduct sensor simulation with it. Tong et al. (2023) adopts similar idea to employ 3D object-level augmentation techniques with NeRF. Different from them, our method does not rely on NeRF, but the point-based neural descriptors, which suffers less from volume rendering artifacts and blurs.

## 3 METHODS

In this section, we firstly give a brief overview of our methods (Section 3.1). Subsequently, we detail the proposed object-level data augmentation approach, which comprises two key steps: reconstruction (Section 3.2) and synthesis(Section 3.3), as shown in Fig. 2.

## 3.1 OVERVIEW

In the context of a specific driving scenario, we have access to several key elements. These include a collection of LiDAR point clouds represented as $\mathbb{P} = \{P_t\}_N$, image set $\mathbb{I} = \{I_t\}_N$, and label sets $\{\mathbb{L}_t\}_N$ corresponding to various timeframes. The parameter $N$ represents the number of frames. Additionally, we hold knowledge of the camera parameters and the ego vehicle's trajectory.

Our goal is to generate high-quality object-level augmented image sets with automatically-generated annotations. To achieve this goal, we propose a novel object-level data augmentation workflow that incorporates point-based scene reconstruction and neural scene rendering, which consists of two key step: reconstruction and synthesis. In the reconstruction step, we firstly separate and accumulate the LiDAR point cloud sequences to obtain denser point cloud for background and objects. Subsequently, we train the renderer and descriptors attached with the accumulated point clouds to reconstruct the scene. In the synthesis step, we firstly position the object accumulated point clouds into the background point cloud. Following this, we apply the same projection and rendering process as employed in the reconstruction step to obtain the edited images and corresponding labels.

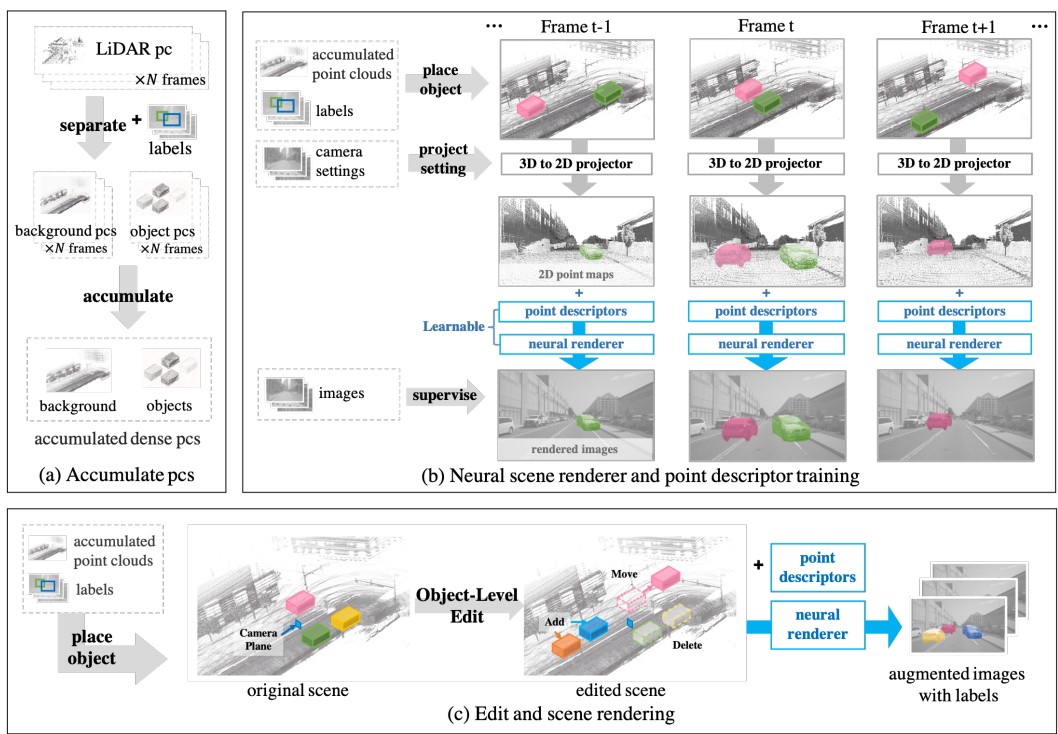

Figure 2: Framework of neural scene rendering based data augmentation. In the reconstruction step, we accumulate point clouds of the background scene and foreground objects (a). The abbreviation 'PC' in the figure stands for 'point cloud.' The information within the scene images is assimilated into the point cloud descriptors and renderer by training (b). In the synthesis step, we perform object-level manipulations on accumulated point clouds, and employ the trained descriptors and renderer to synthesize augmented images and their labels (c).

## 3.2    3D RECONSTRUCTION

### 3.2.1    DENSE POINT CLOUDS GENERATION

In order to enhance the scene reconstruction capabilities of point clouds, we increase their density, enabling them to carry more information for subsequent editing. By separating and accumulating multi-frame LiDAR point clouds using labels and ego's trajectory, we can obtain dense point clouds for the scene and target objects. In a more specific context, at a specific time frame $t$, we can get an individual point cloud denoted as $\boldsymbol{P}_t$. We use labels for all target objects within the frame, represented as $\mathbb{L}_t = \{\boldsymbol{b}, c, id\}_n$, including its bounding box annotation $\boldsymbol{b}$, category $c$, and a unique $id$ to distinguish the object. Consequently, we are able to get a object's the point cloud $\boldsymbol{P}_{id,t}$ using its annotation $\boldsymbol{b}_{id,t}$ and a simple function $f_{inbox}$ to calculate whether a point is inside a bounding box. Besides, we can subtract all objects' point clouds from the original to get the background scene point cloud $\boldsymbol{P}_{bg,t}$:

$$\boldsymbol{P}_{id,t} = f_{inbox}(\boldsymbol{P}_t, \boldsymbol{b}_{id,t}) \qquad \boldsymbol{P}_{bg,t} = \boldsymbol{P}_t - \underset{id}{\cup}\boldsymbol{P}_{id,t} \qquad (1)$$

Upon processing all frame point clouds, we accumulate all background point clouds with the point clouds representing objects. For the background point clouds, we first establish a consistent global coordinate system. Subsequently, guided by the vehicle's trajectory, we calculate the rotation matrix $\boldsymbol{R}_t$ and the translation vector $\boldsymbol{T}_t$ that represent the transformation from the point cloud's coordinate system for frame t to the global coordinate system. Consequently, we apply these transformations and accumulate the corresponding point clouds to obtain the final dense point cloud, denoted as $\boldsymbol{P}_{bg}$. Similarly, we utilize the rotation angle $\boldsymbol{R}_{id,t}$ and the translation vector $\boldsymbol{T}_{id,t}$ (with the corresponding $id$) from the object label to obtain the object point cloud $\boldsymbol{P}_{id}$. :

$$\boldsymbol{P}_{bg} = \underset{id}{\cup}\boldsymbol{R}_t\boldsymbol{T}_t\boldsymbol{P}_{bg,t} \qquad \boldsymbol{P}_{id} = \underset{id}{\cup}\boldsymbol{R}_{id,t}\boldsymbol{T}_{id,t}\boldsymbol{P}_{id,t} \qquad (2)$$

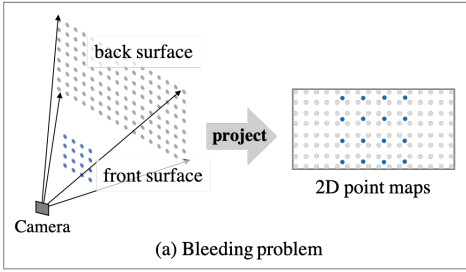 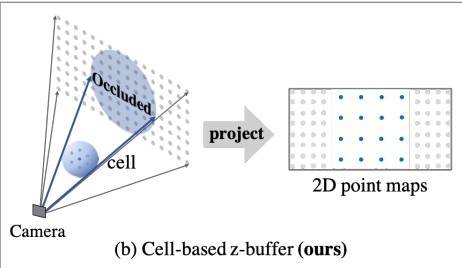

(a) Bleeding problem     (b) Cell-based z-buffer **(ours)**

Figure 3: Bleeding problem (a) and our solution: Cell-based z-buffer (b). We assign every foreground point with a certain physical volume (called *'cell'*), enabling more accurate occlusion calculations and preventing occluded objects being observed by the camera.

### 3.2.2 NEURAL SCENE RENDERER AND DESCRIPTORS TRAINING

After obtaining the dense scene point clouds, the next step is to transform them into a truly informative medium capable of represent the scene for subsequent processing.

Influenced by past point-based methodologies (Aliev et al., 2020; Li et al., 2022b), we introduce two learning components, point descriptors and the neural scene renderer, to make the dense point cloud memorize the scene's information. Each point $p_i$ in the accumulated dense point cloud $P = P_{bg} \cup P_{id}$ is attached with a $M$-dimensional vector as neural descriptor $d_i$, which is used to describe the contextual features of that point and its surroundings within the scene. The neural scene renderer is designed for interpreting the abstraction of point clouds and descriptors to reconstruct the scene, subsequently rendering them into an RGB image. We leverage information from the dataset to train the aforementioned point descriptors and neural renderer. During training, we ensure dense point clouds match real scenes in the dataset. Descriptors and renderer create images from the same viewpoints as ground truth RGB images. Using these RGB images as supervision, descriptors and renderer comprehensively represent the scene. Specifically, our training process unfolds in the following steps (as depicted in Figure 2-b):

**Position foreground objects.** As aforementioned, we have separated the point clouds of background scene and foreground objects. Following the annotations, we position the dense foreground object point clouds back into the scene's point cloud, aligning with the ground truth, denoted as $P_t^{dense}$. In this way, we can unify the descriptors of the same objects across different frames.

**3D-to-2D projection.** To proceed with the subsequent rendering, it is necessary to project the processed points onto the target view plane. More formally, for a specific frame at timestep $t$ with camera setting $c_t$, we project the dense point cloud $P_t^{dense}$ and the descriptors $D = \{d_1, d_2, ..., d_n\}$ onto a specific point map $I_t$, using the projection and raster function $\Phi_{proj}$.

$$I_t = \Phi_{proj}(P_t^{dense}, c_t, D) \qquad (3)$$

For each point $p_i$ which projects to pixel $(x, y)$, we set $I_t[x, y] = d_i$ .

Specifically, we propose a projection algorithm called *cell-based z-buffer* to remove the occluded points and solve the *bleeding problem*(as depicted in Figure 3-a). The bleeding problem is triggered by the lack of topological information in the point cloud. In some case, points from the occluded surfaces can be seen through the front surface. This causes the descriptors to learn incorrect information from the wrong objects. Some methods(Aliev et al., 2020; Li et al., 2022b) solve this problem by progressive learning, which are not suitable for accumulated LiDAR point clouds, because they are far sparser than the ones such methods processed. We introduce a solution *cell-based z-buffer*(as depicted in Figure 3-b) using a more fundamental and direct approach. We assign a certain physical volume to each point to eliminate the no-topological disadvantage. Every point is seen as a cell with a radius $U$. Points behind the cell are treated as obscured points and will not be projected onto the camera plane.

**Rendering and supervision.** We employ an UNet-like architecture (Li et al., 2022b) with learnable parameter $\theta$ as the neural scene renderer $U$ to recover images from the projected point map $I_t$ at time $t$. The rendered RGB image is denoted as $I_{rgb,t}$ :

$$\boldsymbol{I}_{rgb,t} = U_\theta(\boldsymbol{I}_t) \tag{4}$$

We use the RGB image $\boldsymbol{I}_{gt,t}$ with the same time $t$ and same camera setting $\boldsymbol{c}_t$ as the ground truth, to supervise the point descriptors and neural renderer training. We employ Huber loss and VGG loss (Johnson et al., 2016) for supervision.

$$L(D, \theta, t) = L_{Huber}(\boldsymbol{I}_{gt,t}, \boldsymbol{I}_{rgb,t}) + L_{VGG}(\boldsymbol{I}_{gt,t}, \boldsymbol{I}_{rgb,t}) \tag{5}$$

### 3.3 OBJECT LEVEL EDITING AND NEW DATA SYNTHESIS

After completing the training of the descriptor and renderer, we can utilize the point cloud as a middleware to achieve object-level editing and data augmentation. In particular, for a specific time frame $t$, we reposition the accumulated point clouds representing objects within the scene according to custom specifications. Given the original accumulated point cloud of the background scene $\boldsymbol{P}_{bg}$ and objects $\mathbb{P}_{obj} = \{\boldsymbol{P}_{id}\}_n$, we can get the new edited accumulated point cloud $\boldsymbol{P}_t^{ed}$. Subsequently, we apply the same projection function $\Phi_{proj}$ as employed in the training step, to project the edited point cloud with its corresponding descriptors $\boldsymbol{D}$ and same camera setting $\boldsymbol{c}_t$. Then we use the trained $U_\theta$ to obtain the edited images $\boldsymbol{I}_t^{ed}$. Simultaneously, the newly customized object information is also preserved as labels $\boldsymbol{L}_t^{ed}$ for the augmented data.

Additionally, we further optimized the generated images to enhance augmented data's quality. When the camera settings remain unchanged, after editing, only a portion of the image is actually affected by the edits. Consequently, we retained the unchanged pixels from the high-quality original image, while for the altered pixels, we integrated the corresponding regions generated by the renderer.

## 4 EXPERIMENTS

### 4.1 DATASET AND METRICS

**Dataset.** We create the augmented data and train the detection model based on nuScenes Caesar et al. (2020) dataset. nuScenes collected 1000 driving scenes, and each scene has 20 second video frames and is annotated around 40 key frames, along with the ego's trajectory. Each of the annotated frames contains six monocular camera images with 360-degree FoV and a 32-beam LiDAR scan. Annotation includes 23 object classes with accurate 3D bounding boxes.

**Metrics.** To demonstrate the efficiency of augmentation, we present official evaluation metrics, including the nuScenes Detection Score (NDS), mean Average Precision (mAP), and five True Positive (TP) metrics: mean Average Translation Error (mATE), mean Average Scale Error (mASE), mean Average Orientation Error (mAOE), mean Average Velocity Error (mAVE), and mean Average Attribute Error (mAAE). Furthermore, for certain ablation studies related to image generation quality, we utilize Peak Signal-to-Noise Ratio (PSNR) and perceptual loss (VGG loss) as evaluation metrics to confirm the enhancements in image generation quality achieved by our designs.

### 4.2 IMPLEMENTATION DETAILS

**Reconstruction setting.** Out of 700 scenes in the nuscenes training set, we randomly selected 160 scenes for reconstruction. During the process of dense point cloud generation, to prevent an excessive concentration of points within a single voxel unit, we applied a sparsification process to the merged point cloud. We retain at most one point within a 0.03m cubic unit. We set the cell radius $r$ of the cell-based z-buffer to be 0.03m, aligning with the point cloud sparsification parameter mentioned above. The dimensionality $M$ of the point descriptors $\boldsymbol{D}$ is set to 8. The specific parameter details for the Neural renderer are consistent with Li et al. (2022b). For each scene, we conduct training for 60 epochs to train the renderer and descriptor.

**Editing and synthesis setting.** We ensure that the composition of objects within each edited scene adheres to the physical laws governing the real world. In the synthesized scenes, occurrences such as object overlap, instantaneous movement, abrupt appearance or disappearance, which defy common sense, are rigorously prevented. To achieve this goal, we initially employed some simple rules for

automated data generation. Subsequently, we manually removed unreasonable portions. Additionally, to guarantee that the data distribution of augmented objects remains consistent with the original dataset, the selection of each edited object is randomized.

## 4.3 AUGMENTATION PROFORMANCE

We employ two influencing mult-view 3D perception model, PETRv2 (Liu et al., 2022b) and BEVFormer (Li et al., 2022a) as our baseline, to investigate the efficiency of proposed augmentation method. Due to computational constraints, we conducted experiments using the most lightweight settings from two baseline models. For PETRv2, we take VoVNet-99 (Lee & Park, 2020) as image backbone using the resolution of $800 \times 320$. For BEVFormer, we take ResNet-50 He et al. (2016) as image backbone using the resolution of $800 \times 450$. The PETR experiments utilized four NVIDIA GTX 3080Ti GPUs total the total batch size of 8, while the BEVFormer experiments employed eight of the same GPUs with the total batch size of 16. We use all the 700 scenes in the nuScenes training set, along with different composition of augmentation sets to train the detectors, and evaluate the detectors on the entire validation set. In particular, we implement a cut-paste Dwibedi et al. (2017) algorithm for multi-view 3D detection for comparison. The relevant implementation details can be found in the appendix A.1. For specific results, please refer to Table 1 and Table 2.

Table 1: Multi-view 3D detection results on nuScenes validation set based on PETRv2.

| Method | mAP↑ | NDS↑ | mATE↓ | mASE↓ | mAOE↓ | mAVE↓ | mAEE↓ |
|---|---|---|---|---|---|---|---|
| Baseline | 40.65 | 50.16 | 73.48 | 26.76 | 44.03 | 38.72 | 18.66 |
| + cut-paste | 36.66 | 43.55 | 79.71 | 27.42 | 54.77 | 65.34 | 20.63 |
| + Add | 40.84 | 50.19 | 73.05 | 27.22 | 46.04 | 37.06 | 18.97 |
| + Move | 40.72 | 50.07 | 72.74 | 27.01 | 45.63 | 38.70 | 18.82 |
| + Delete | 40.68 | 50.34 | 72.15 | 26.68 | 44.88 | 37.56 | 18.72 |
| + Add & Delete | 41.09 | 50.05 | 71.75 | 26.98 | 47.54 | 39.63 | 19.10 |
| + Add & Move | 41.11 | 50.28 | 72.26 | 27.02 | 45.88 | 39.26 | 18.31 |
| + Move & Delete | 41.01 | 50.46 | 71.52 | 27.05 | 45.30 | 37.82 | 18.76 |
| + All | **41.45** | **50.64** | 71.89 | 26.87 | 44.84 | 38.26 | 18.90 |

Table 2: Multi-view 3D detection results on nuScenes validation set based on BEVFormer.

| Method | mAP↑ | NDS↑ | mATE↓ | mASE↓ | mAOE↓ | mAVE↓ | mAEE↓ |
|---|---|---|---|---|---|---|---|
| Baseline | 25.24 | 35.40 | 89.98 | 29.38 | 65.52 | 65.67 | 21.61 |
| + cut-paste | 23.54 | 33.34 | 95.79 | 28.86 | 68.26 | 68.93 | 22.42 |
| + ours | **26.13** | **35.88** | 89.37 | 29.18 | 64.30 | 66.56 | 22.47 |

As shown in Tab. 1, we first compare the performance with PETRv2 baseline and cut-paste augmented detector, along with different compositions of our augmented data. Compared with baseline, our method improves mAP by $0.80\%$ and NDS by $0.48\%$. Besides, proposed method also get $0.89\%$ mAP and $0.44\%$ NDS promoted on BEVFormer(as shown in Tab. 2). These consistent improvements indicate that proposed augmentation algorithm can bring stable improvement with different baselines. In the context of the cut-paste method, due to its handling issues like occlusion computation and temporal consistency, often leads to lower augmented data quality and consequently reduced model performance. Our approach effectively overcomes these challenges, resulting in higher-quality synthetic data for the entire scene and consequently improves model performance.

## 4.4 ABLATION STUDY

### 4.4.1 POINT CLOUD SIZE

In order to investigate the influence of accumulated point cloud size on scene reconstruction capability, we selected the sparsified cube size (as mentioned in Section 4.2) as the parameter controlling point cloud density and divided it into three levels: 0.02m, 0.03m, and 0.04m for our study. For each level, we randomly selected ten scenes from the nuScenes dataset for reconstruction, using the quality of generated rendered images as the metric for reconstruction quality assessment. The results are presented in Tab. 3. The data demonstrates that an increase in point cloud density correlates with a certain improvement in data generation quality. We opt for a 0.03m sparsified cube size to strike a balance between model parameters and marginal improvements in image quality.

Table 3: Image generation quality with different pointcloud size.

| Sparsified cube size | PSNR↑ | VGG↓ | Parameters |
|---|---|---|---|
| 0.02m | 26.57 | 322.85 | 21.13M |
| 0.03m | 26.44 | 321.41 | 17.43M |
| 0.04m | 25.64 | 520.63 | 7.66M |

### 4.4.2 CELL-BASED Z-BUFFER

In order to evaluate the efficiency of proposed cell-based z-buffer in Section 3.2.2, we conduct experiments on 10 scenes from the nuScenes v1.0-mini dataset, comparing the image generation quality using different z-buffer settings. The test results is presented in the Tab. 4 (for more detailed information, please refer to the appendix A.2). The data shows a remarkable improvement in image quality with cell-based z-buffer usage. In addition, we conducts a comparative visualization between the two approaches (as depicted in Fig. 4-a). The utilization of the original z-buffer method encounters difficulties in effectively reconstructing the foreground vehicle due to the bleeding problem. In contrast, our method successfully addresses this challenge.

Table 4: Image generation quality with different z-buffer settings.

| Method | PSNR↑ | VGG↓ |
|---|---|---|
| orignal z-buffer | 19.56 | 680.38 |
| cell-based z-buffer (ours) | 25.23 | 350.22 |

## 4.5 VISUALIZATIONS

Fig. 4 and Appendix A.3 present some example results of different editing cases. In cases b, e, and f, we demonstrate the method's capacity to synthesize novel views of objects through addition and relocation. In cases c and d, the method excels at object removal and pixel filling, showcasing it excellent background inpainting capabilities. The operations in cases c and e involve multiple views, illustrating the method's proficiency in preserving scene consistency when generating multi-view data.

## 5 CONCLUSION

In this paper, we introduce a novel approach for object-level data augmentation in the context of visual autonomous driving. This approach combines scene reconstruction and neural scene rendering, effectively addresses challenges related to 2D/3D inconsistencies. By enhancing multi-camera the performance of multi-camera detectors and providing realistic training data, our method enriches data diversity and consistently improves model performance

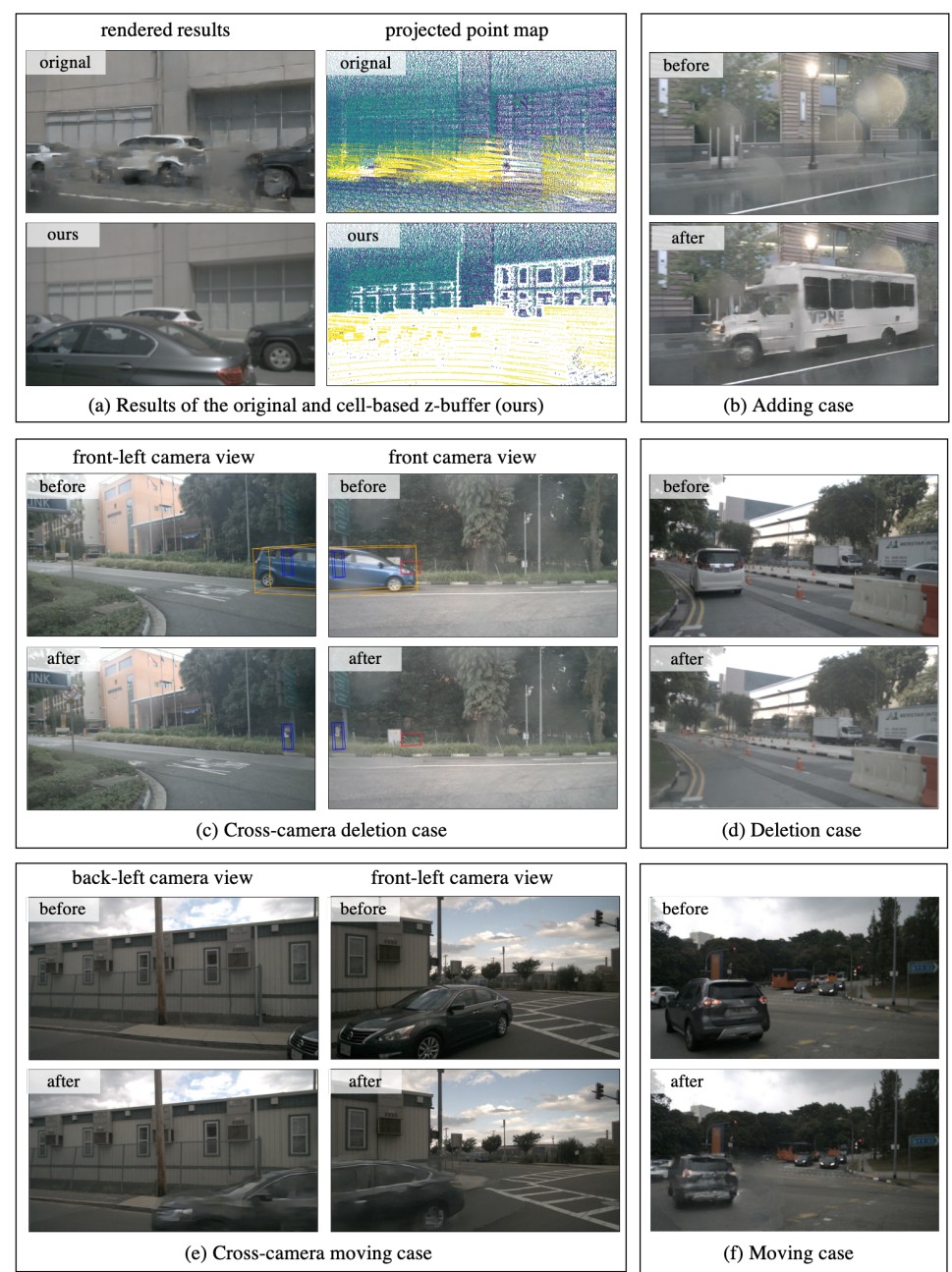

Figure 4: Visualization of z-buffers (a) and example results of different editing cases (b-f). Figure (a) presents the effectiveness of proposed cell-based z-buffer to solve the bleeding problem and reconstruct the foreground objects. Figure (b-f) demonstrates the method's ability to address the cross-camera issue (c,e), novel view synthesis issue (b,e,f), background inpainting issue (c,d).

Our task holds the potential for expansion and scalability. It can be extended to address other tasks, such as segmentation and occupancy estimation. Additionally, there is potential to integrate lidar perception for multimodal enhancement. However, current solution has certain limitations. Firstly, the sparse nature of distant points leads to poor reconstruction appearance in remote regions. Secondly, our method is limited to reconstructing objects from specific angles which appears in the dataset. Thirdly, the method's accuracy is compromised due to the 'bleeding problem.' Addressing these challenges will be crucial for future advancements in our approach.

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

# A APPENDIX

## A.1 CUT-PASTE DATA AUGMENTATION IN MULTI-VIEW 3D DETECTION

We implemented a cut-paste data augmentation algorithm (Dwibedi et al., 2017) tailored for the multi-view 3D detection task as a naive object-level augmentation method, which is used for comparison with the proposed method in section 4.3. The workflow is shown in Fig. A.1.

The algorithm augment the target sample with the cut object bounding boxes form the source sample. Notably, when selecting objects from the source sample, we consider their visibility in the source images (referred to as the Filter-A process in the figure). We choose objects with higher visibility for pasting. When pasting the boxes onto the target scene, we paste the boxes to the position same as their position in the source images. We also consider the occlusion caused by the pasted objects. If the pasted box would heavily occlude the original scene objects, we will not select box to be pasted (referred to as the Filter-B process in the figure).

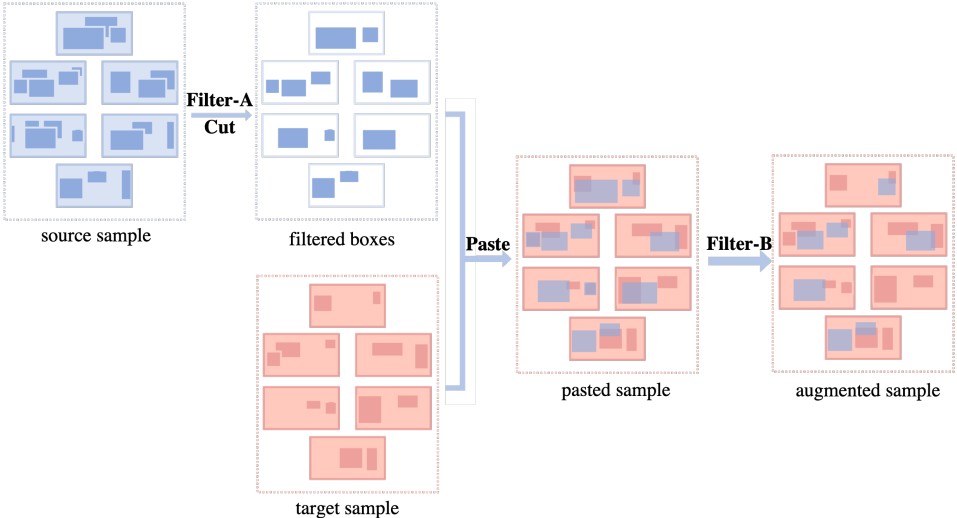

Figure 5: Workflow of cut-paste data augmentation algorithm in multi-view 3D detection.

## A.2 DETAILS OF Z-BUFFER ABULATION STUDY

We conduct experiments on 10 scenes from the nuScenes v1.0-mini dataset. For the reconstruction of each scene, we extracted 90% of the images for use as the training dataset and allocated the rest 10% images for the validation dataset. The following are the detailed quantitative results obtained by testing each scene on its respective validation set.

## A.3 MORE VISUALIZATIONS

Here we list some visualization of augmented data with annotations.

Table 5: Image generation quality with different z-buffer setting.

| Scene | VGG | PSNR | VGG | PSNR |
|---|---|---|---|---|
| scene-0061 | 666.91 | 18.52 | 293.09 | 27.42 |
| scene-0103 | 616.28 | 18.46 | 269.85 | 26.56 |
| scene-0553 | 261.96 | 26.84 | 230.70 | 27.82 |
| scene-0655 | 587.34 | 18.80 | 227.77 | 29.77 |
| scene-0757 | 430.29 | 21.77 | 739.63 | 15.60 |
| scene-0796 | 842.73 | 16.44 | 350.51 | 26.18 |
| scene-0916 | 751.07 | 16.98 | 278.13 | 27.81 |
| scene-1077 | 738.06 | 20.98 | 355.96 | 25.11 |
| scene-1094 | 894.63 | 18.90 | 385.30 | 24.32 |
| scene-1100 | 1014.57 | 17.87 | 371.30 | 24.68 |
| Average | 680.38 | 19.56 | 350.22 | 25.23 |

**Before**                                                          **After**

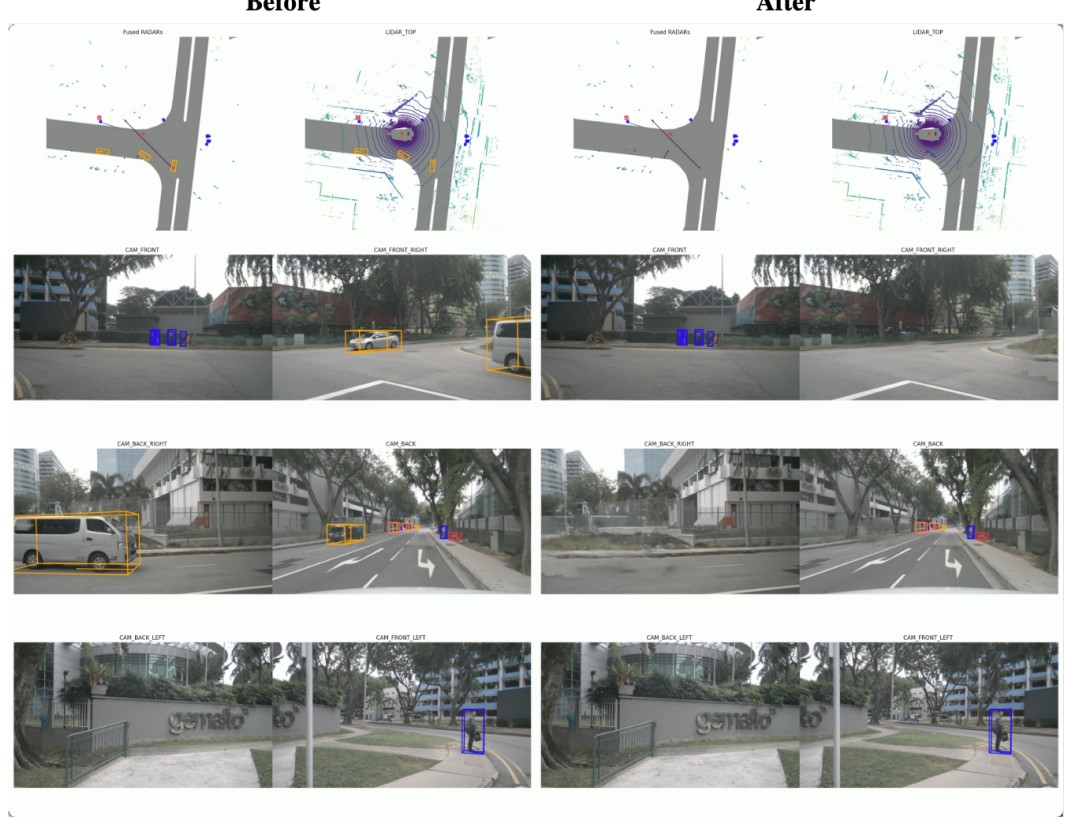

Figure 6: Examples with deleting moving vehicles. Visualized along with annotations.

**Before**                    **After**

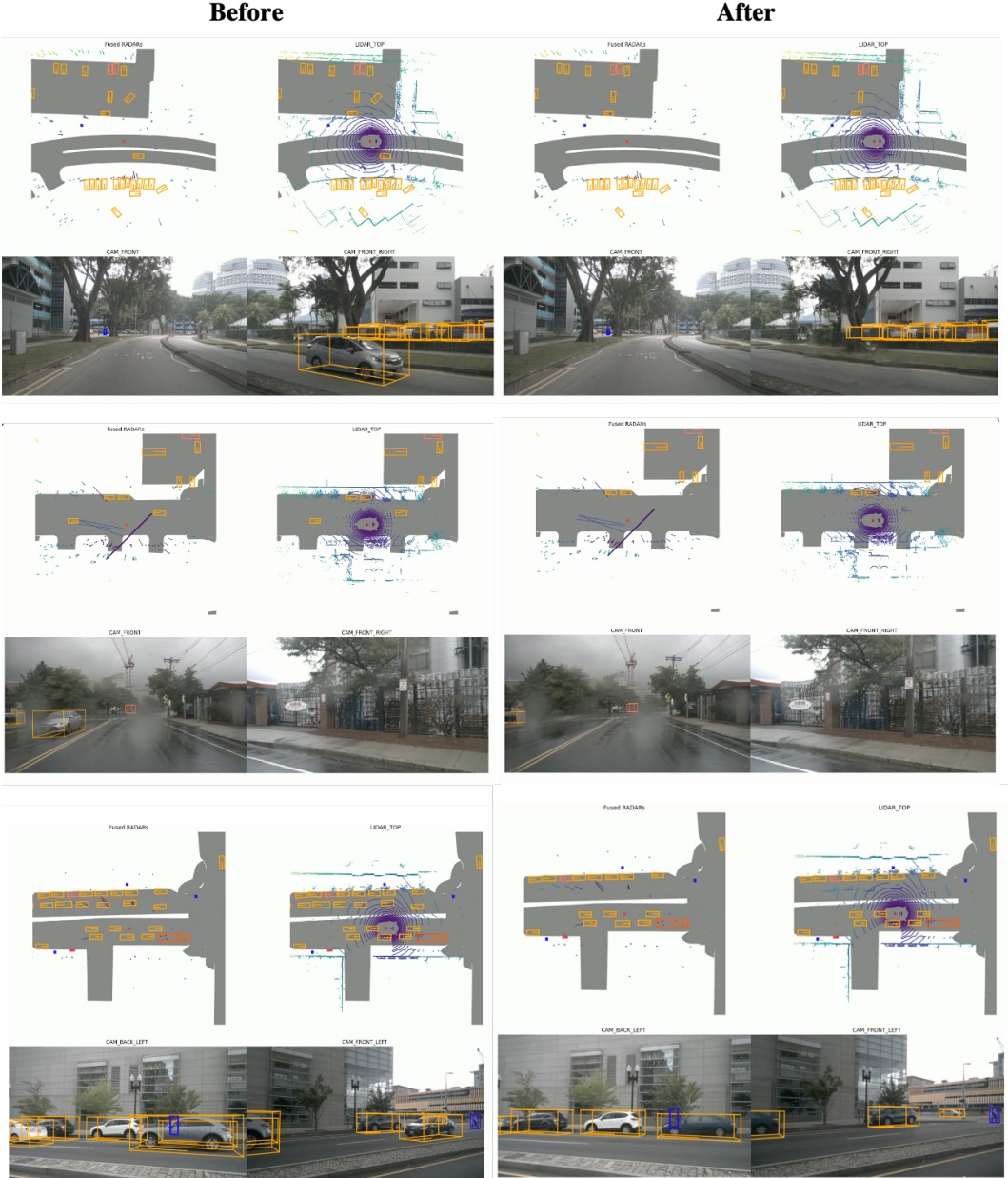

Figure 7: Examples with deleting moving vehicles. Visualized along with annotations.

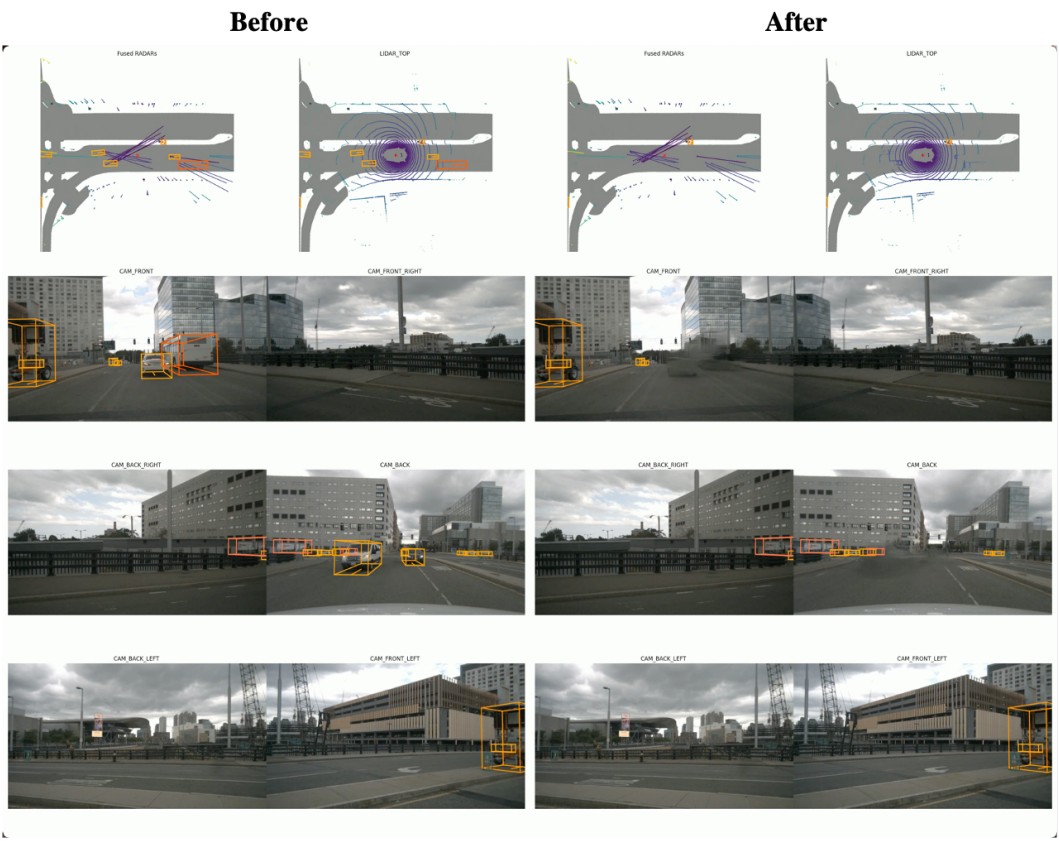

Figure 8: Examples with editing static vehicles. Visualized along with annotations.

