# OpenReview forum: "Object-level Data Augmentation for Visual 3D Object Detection in Autonomous Driving"
_ICLR.cc/2024/Conference — Submitted to ICLR 2024_

### Official Review · Reviewer_v2XH · 2023-10-28

**Soundness:** 4 excellent
**Presentation:** 4 excellent
**Contribution:** 3 good
**Rating:** 6
**Confidence:** 5

**Summary:**

In this paper, the authors try to address insufficient scene diversity for visual 3D object detection in autonomous driving. They propose an object-level data augmentation workflow, by reconstruction and synthesis. To do this,  they split and accumulated the foreground and background of the global pointcloud first. Then, they train the learnable point descriptors and the neural renderer to recover the original scene image. In the synthesis step, they perform object-level transformations on accumulated point clouds and utilize the trained renderer to synthesize augmented images and their labels.

**Strengths:**

The authors challenge the core problem of machine learning, i.e., the insufficient diversity of data samples. They try to generate more data samples by reconstruction and synthesis. They propose a *cell-based z-buffer* algorithm to remove the occluded points and solve the *bleeding problem*.  I think the proposed solution is simple and straightforward. Thus, I think the proposed approach is easily reproducible.

**Weaknesses:**

I have some concerns about the proposed method and experiment result:
1. Is there any safeguard to guarantee the augmented data is located in reasonable regions?
2. How does the algorithm control the appearance style of the added object? Is there a way to generate an unseen appearance from the training data set?
3. The PSNR of the synthesis image is <30 dB, which means the human eye distinguishes the synthesis image from the original image. It raises a question in my mind: does the quality of the synthesis data affect downstream tasks? An experiment that compares the results trained within the original image and the synthesis data is appreciated.

**Questions:**

See the weakness.

---

> ### Author Response · Authors · 2023-11-17
> **Response to Reviewer v2XH**
>
> We are grateful for the thorough review you provided for our manuscript, as well as for your positive feedback on our research efforts. We address the ambiguities in our approach and incorporate additional experiments to respond to your queries.
>
> **Q1. How to guarantee the edited object be located in reasonable regions?**
>
> We design a set of detailed rules for batch object editing, which could ensure that the edited objects appear in reasonable positions:
>
> - **Adding objects**:
>   - In driving scenarios, objects are mostly located on the ground plane. Therefore, we tend to place the added object on the ground. To achieve this, we approximate the ground plane function by analyzing scene data, including point cloud and object bounding boxes.
>   - We randomly select a point within a certain range on the ground plane, place the object at that position, and rotate the object by a random angle parallel to the ground plane.
>   - Subsequently, we perform collision detection between the added object's bounding box and other objects or the background scene. If there is a physical overlap with other objects, we will attempt another placement.
> - **Moving objects**:
>   - When moving static objects, we randomly choose a direction parallel to the ground plane and move the object a certain distance.
>   - When moving dynamic objects, because of tricky issues such as velocity and direction, we employ a simple strategy: For each edit, we randomly select a number x. When handling a certain moving object at time t, we move it to its position at time t-x.
>   - After an object is moved, we conduct collision calculations. If there is a physical overlap, the object undergoes re-editing.
> - **Deleting objects**:
>   - We randomly select an object in the scene for deletion. Most of the time the deletion operation does not result in the violation of real-world physical laws.
>
> Finally, we perform manual inspections of the edited scenes to eliminate implausible elements.
>
> This part will be included in the final version of the paper.
>
> **Q2. How to control the appearance style of the edited object? Is there a way to generate an unseen appearance from the training data set?**
>
> The proposed augmentation method is based on the reconstruction of scenes and objects. Consequently, their appearance style is constrained to those present in the training dataset. The proposed method cannot generate object appearances with an unseen style.
>
> Recently, we have come across some excellent works about synthesizing new appearance styles of objects. It is an interesting and potential idea to train a unified generator for the objects in the same class, which could learn their common features and control the generated appearance style. In future work, we will explore this idea.
>
> **Q3. Comparison of the training results within the original image and the synthesis data.**
>
> We design an experiment to discuss this issue. Considering the limited resources, we use 20% of the nuScenes data (a total of 140 scenes) as a mini-dataset for training. Additionally, we select 28 scenes from the mini-dataset and process them using different synthesis methods, which are then added to the mini-dataset. The synthesis methods we employed are as follows:
>
> - **Rerender:** We rerender the original images by employing the same reconstruct & render method of the proposed method, without any camera setting and scene layout changes.
> - **Original:** We make no modifications to the original images and add them directly into the mini dataset. This method serves as a baseline.
>
> We employ these different mini datasets to train PETRv2, and the detailed training settings align with those outlined in Section 4.3 of the original paper. The detailed performance results are as follows.
>
> | Dataset settings                                     | Synthesis Method | Synthesis data PSNR | Detection    mAP | Detection  NDS |
> | ---------------------------------------------------- | ---------------- | ------------------- | ---------------- | -------------- |
> | 140 original scene data + 28 **original** scene data | /                | /                   | 33.06            | 39.89          |
> | 140 original scenes + 28 **synthesis** scene data    | Rerender         | 26.3*               | 32.58            | 40.37          |
>
> [*] Note: We randomly sample a certain number of generated images, compare them to the original images, and calculate this PSNR value.
>
> While there is indeed a gap in the quality between synthetic and original data, experimental results indicate that these differences do not significantly impact the model's detection performance. This further indicates that the proposed data generation method is applicable to downstream tasks.

---

> > ### Comment · Reviewer_v2XH · 2023-11-21
> >
> > Thanks for your rebuttal.
> >
> > ***R1.*** The authors design a semi-automatic annotation pipeline to process the augmented data and generate the deformable objects that appear in reasonable regions. Is there a cerite on whether to manually refine the augmented data? Is it removed if augmented data's translation, rotation, and scale are biased in rendered visual data?
> >
> > ***R2.*** Thanks. It solved my concern.
> >
> > ***R3.*** Thank you. It is an exciting finding. How about the extreme case, i.e., pure synthesis data and pure original data? What is the impact of the ratio between the original data and the synthesis data in downstream tasks?

---

> > > ### Author Response · Authors · 2023-11-22
> > > **Response to Reviewer v2XH (part 2)**
> > >
> > > **Reply to R1.**
> > >
> > > - Actually, automated editing rules can cover most editing scenarios, generating rational and effective data. Failures may occur in a very small number of instances. The following are two examples:
> > >
> > >   - In nuScenes scene-570, there are wire fences on both sides of the road. However, LiDAR point clouds struggle to capture such hollow structures. Therefore, during collision detection, the volume of the wire fence is ignored, resulting in some edited objects being mistakenly placed across the wire fence.
> > >   - In nuScenes scene-790, there is a high platform on the left side of the road. However, our algorithm approximates the ground plane aligning with the road plane. Consequently, some edited objects, when placed according to the fitted ground plane, are mistakenly positioned inside the high platform entity.
> > >
> > >   We manually remove these failures from the synthesis dataset.
> > >
> > > - **About translation/rotation/scale bias.** Here we emphasize that our editing method maintains consistency between 2D and 3D. The edited results align precisely with the expected positions, shapes, and sizes. Therefore, no translation/rotation/scale bias will be generated during the synthesis process. Our manual operations only include the illogical failures mentioned above.
> > >
> > >
> > >
> > > **Reply to R3.**
> > >
> > > We conduct several additional experiments to address your question.
> > >
> > > For a quick evaluation, we use 5k keyframes of the nuScenes data as a mini-dataset for training, and replace different percentages of the mini-dataset with synthetic data. We employ these different mini datasets to train PETRv2, and the detailed training settings align with those outlined in Section 4.3 of the original paper. The performance results are as follows.
> > >
> > > | Methods                                | mAP       | NDS       | mATE  | mASE  | mAOE  | mAVE   | mAAE  |
> > > | -------------------------------------- | --------- | --------- | ----- | ----- | ----- | ------ | ----- |
> > > | 100% original data                     | **26.75** | **31.83** | 91.40 | 29.87 | 82.51 | 88.23  | 23.43 |
> > > | 50% original data + 50% synthesis data | 26.40     | 29.83     | 90.81 | 29.82 | 83.46 | 102.17 | 29.65 |
> > > | 100% synthesis data                    | 25.40     | 29.35     | 93.29 | 30.46 | 83.45 | 99.42  | 26.86 |
> > >
> > > We find that the detection performance of the model gradually declines with the increase of the proportion of synthetic data. This shows that augmented synthetic data can play an auxiliary role in model training, but can not replace the original data.
> > >
> > > Considering the limited time resources, we can only provide results of very limited ratio settings. We will present more detailed experiments and analyses in the final version.

---

> > > > ### Comment · Reviewer_v2XH · 2023-11-22
> > > >
> > > > Thanks for the answers. It solved my concerns. I tend to raise my rating.

---

> > > > > ### Author Response · Authors · 2023-11-23
> > > > >
> > > > > Thanks for your reply! We are glad that our responses solve your concerns. We kindly remind you that you may need to **update the score accordingly**.

---

### Official Review · Reviewer_m1gL · 2023-10-30

**Soundness:** 3 good
**Presentation:** 3 good
**Contribution:** 3 good
**Rating:** 5
**Confidence:** 5

**Summary:**

Visual 3D object detection is an important task in the 3D perception system. However, labeling 3D data is extremely expensive and the diversity of training data plays an important role in deep learning. To this end, this work proposes a rendering-based instance-level data augmentation method for image-based 3D detection. In particular, this method extracts image features from sequences and aligns them with associated LiDAR point clouds, and augments the objects in the 3D space. After that, a neural scene renderer is adapted to generate the 2D images with the augmented objects. The experiments on the nuScenes with two base detectors show the effectiveness of the proposed data augmentation strategy.

**Strengths:**

- The work is well-motivated. The diversity of training data plays an important role in deep learning, and conducting data augmentation is really hard for 3D detection due to the geometric consistency. Therefore, exploring effective 3D data augmentation methods is a meaningful topic.
- The experiments on nuScenes dataset show the effectiveness of the proposed method.

**Weaknesses:**

- The overall data-augmentation process is complex, involving multiple additional data, such as point cloud, sample sequences, external renders, etc. Considering this, performance improvement is limited, maybe evaluating a smaller dataset, such as KITTI,  can better show the effectiveness of this data augmentation method.

- Besides, the computation cost will be inevitably increased. The cost of generating the new training samples should be reported.

- The authors only report the results on the nuScenes validation set, and the test set should also be presented.

- ROI-10D [1] also provides an instance-level data augmentation method for image-based 3D detection. Discussion and comparison should be included.

References:
[1] ROI-10D: Monocular Lifting of 2D Detection to 6D Pose and Metric Shape, CVPR'19 (also see the supp)

**Questions:**

See weaknesses.

---

> ### Author Response · Authors · 2023-11-17
> **Response to Reviewer m1gL (part 1)**
>
> We sincerely appreciate your feedback and suggestions. We are greatly encouraged that you found our method is *'well-motivated'* and *'effective'*. We conduct extensive studies and added experimental evaluations to comprehensively address your concerns.
>
> **Q1. Evaluation on a smaller dataset.**
>
> Thank you for your suggestions. We conduct validation on a smaller dataset.
>
> - We have successfully implemented the scene reconstruction method on the KITTI dataset. **Visualization can be found at anonymous URL** https://anonymous.4open.science/r/kittivisualization/kittivis.png
>
> - Unfortunately, due to inherent limitations in the KITTI dataset, further editing of the reconstructed scenes is not feasible.
>
>   The proposed augmentation method requires the dataset to provide two types of labels: the **driving trajectory** of the ego and the **bounding boxes** of objects. Regrettably, sequences in the KITTI dataset often do not concurrently possess both of these labels.
>
>   Specifically, in the KITTI Visual Odometry dataset, a subset of sequences from the original KITTI raw dataset is annotated with vehicle trajectories, while another subset in the KITTI detection dataset is annotated with object bounding boxes. Notably, there is no overlap between the sets of sequences annotated for these two types of labels.
>
> Therefore, to facilitate validation on a smaller dataset, we adopt an alternative approach:
>
> - We randomly sampled 20% of the nuScenes training set data to construct a small-scale dataset (including 140 scenes).
> - Within this small-scale dataset, we further selected 20% of the data for augmentation (including 28 scenes) using the proposed augmentation method.
> - Lastly, we employed the PETRv2 model (training settings consistent with those outlined in Section 4.3 of the paper) to train on both the augmented and non-augmented datasets, subsequently conducting tests on the validation set. Following is the performance data:
>
> |              | mAP↑      | NDS↑      | mATE↓     | mASE↓     | mAOE↓     | mAVE↓     | mAAE↓     |
> | ------------ | --------- | --------- | --------- | --------- | --------- | --------- | --------- |
> | **Baseline** | 32.96     | 39.40     | 87.40     | 28.69     | 64.68     | 67.95     | 22.10     |
> | + **Ours**   | **33.27** | **41.18** | **84.47** | **28.07** | **60.06** | **61.34** | **20.63** |
>
> It can be observed that our method exhibits consistent performance in the small-scale dataset.
>
> **Q2. Data generation cost.**
>
> Please note that our data synthesis process is **offline**. We generate the augmented data and store them before the detection model training. During the training process, we directly use the stored synthetic data. Therefore, the proposed augmentation process does not affect the training efficiency of the detection model.
>
> We also provide the detailed computation costs of generating the new training samples as below:
>
> - In the reconstruction stage：
>   - **Time cost:** Our descriptors and render are trained on 4 RTX 3080Ti (12G) GPUS, which takes about **5 hours** for a scene with about 240 images (with a resolution of 1600x900).
>   - **Memory cost:** For a single scene, trained point-cloud descriptors take about **50MB~200MB** of storage, depending on the size of the point cloud. The trained UNet takes **116MB** as the renderer.
>
> - In the rendering stage, the trained renderer takes about **20 seconds** to render a single image (with a resolution of 1600x900) from the point map on a single RTX 3080Ti (12G) GPU.
>
> **Q3. Results in nuScenes test set.**
>
> We conduct testing on the test set, and submit the results to the nuScenes detection challenge, obtaining the following performance metrics:
>
> | Method             | mAP↑      | NDS↑      | mATE↓     | mASE↓     | mAOE↓     | mAVE↓     | mAAE↓     |
> | ------------------ | --------- | --------- | --------- | --------- | --------- | --------- | --------- |
> | **PETRv2***        | 39.68     | 49.39     | 69.56     | 26.27     | 47.49     | 48.87     | 12.36     |
> | **PETRv2* + ours** | **42.00** | **51.27** | **67.05** | **25.97** | **47.29** | **44.81** | **12.15** |
>
> The results of the test set further demonstrate that our method has brought significant performance improvements to the baseline.
>
> *Note： The best device we can acquire is RTX 3080Ti (12G), which is hard to support large image resolution and strong backbones. Therefore, we use a training setting consistent with the paper's Section 4.3: a lightweight backbone and lower-resolution inputs. Additionally, we only utilize the training set for training and augmentation. As a result, the data in the table differs significantly from models at the forefront of the challenge leaderboard, which employ heavy backbones, high-resolution image inputs, and train on the entire dataset (train+val).

---

> ### Author Response · Authors · 2023-11-17
> **Response to Reviewer m1gL (part 2)**
>
> **Q4. Discussion and comparison with ROI-10D.**
>
> RoI-10D focuses on monocular 3D detection, which also provides certain methods to synthesize object-level edited images. In comparison to our method, the differences are outlined as follows:
>
> - ROI-10D relies on CAD vehicle models to extract the vehicle meshes. Consequently, the range of augmented object categories and shapes is restricted. Our method, however, allows for the editing of all target objects present in the dataset, from small traffic cones to huge trailers.
> - ROI-10D employs a single-frame image as input to reconstruct vehicles into 3D textured meshes. In contrast, our method utilizes the images and point cloud sequence across multiple frames, enabling the learning of more precise and detailed object shapes and texture information.
> - ROI-10D's object editing methods are limited. For example, it cannot handle the removal of existing objects in the scene. Our method supports more flexible editing operations, including free rotation, movement, and deletion.
>
> Regarding quantitative performance comparison, RoI-10D does not provide open-source code. Given the limited time window for the rebuttal phase, we regret that we are unable to reproduce it and give a quantitative assessment. We will complete the relevant experiments and present the results in the final version of the paper.

---

> ### Author Response · Authors · 2023-11-22
>
> We hope these supplementary experiments and discussions can address your concerns. Please let us know if there are any additional clarifications or experiments that we can offer. We would love to discuss more if any concern still remains.

---

### Official Review · Reviewer_ZS78 · 2023-10-31

**Soundness:** 3 good
**Presentation:** 3 good
**Contribution:** 3 good
**Rating:** 6
**Confidence:** 3

**Summary:**

The authors explore and analyze the existing data augmentation methods for 3D object detection, and propose to adopt scene reconstruction and neural scene rendering to scale up the dataset. The experimental results on nuScenes, show the effectiveness of the proposed method.

**Strengths:**

The task of data augmentation for 3D object detection is popular and interesting in the 3D community. The authors propose to use scene reconstruction and neural scene rendering to enlarge the dataset and make it work on nuScenes.

**Weaknesses:**

1. Comparison with data augmentation SOTAs. Although the authors compared the proposed method with the vanilla Cut & Paste in terms of the performance. However, since Cut & Paste doesn't use any point cloud raw data, so to have a fair comparison, it would be interesting if the authors could show the performance comparison with LiDAR-based data augmentation methods, i.e., Tongetal.(2023).

2. It is unclear to me how large the augmented dataset is compared with the original one.

3. It would be interesting if the authors could show the detailed detection performance of 10 classes on the nuScenes. From my understanding, the proposed method is kind of sensitive to different objects with different sizes. Since the small-scale objects, like pedestrians and cyclists might only take a couple of pixels.

**Questions:**

Please refer to the weakness part.

---

> ### Author Response · Authors · 2023-11-17
> **Response to Reviewer ZS78**
>
> We express our gratitude for your valuable feedback and suggestions, which serve as a source of great encouragement to us. While there are still some concerns and questions, we are doing our best to address them.
>
> **Q1. Comparison with other methods.**
>
> Firstly, LiDAR-based augmentation methods are primarily employed for LiDAR 3D perception. Augmentation in the LiDAR point cloud is hard to synchronize to the image level. Our approach utilizes LiDAR as an intermediary to achieve augmentation of the image. In essence, our method serves as a bridge between LiDAR-based and image augmentation.
>
> Secondly, Tong et al. (2023) mentioned does not necessarily rely on LiDAR data. In this method, LiDAR serves as a supplementary depth information source for NeRF procession. As Tong et al. (2023) do not public their source code and disclose detailed editing settings, within the limited rebuttal time windows, we apologize that we cannot conduct a quantitative comparison of augmentation algorithm performance. We will complete the relevant experiments and present the results in the final version of the paper.
>
> **Q2. Data size comparison of original and augmented data.**
>
> The original nuScenes training dataset comprised **700 scenes**. We randomly selected **20%** of these scenes for augmentation. In the final training dataset, the ratio between original data and generated data was maintained at **5:1**.
>
> **Q3. Augmented performance of different classes of objects.**
>
> We provide the detailed detection performance data of each object in nuScenes of PETRv2 in the following table.
>
> | Methods      | mAP       | car       | truck     | bus       | trailer   | construction_vehicle | pedestrian | motorcycle | bicycle   | traffic_cone | barrier   |
> | ------------ | --------- | --------- | --------- | --------- | --------- | -------------------- | ---------- | ---------- | --------- | ------------ | --------- |
> | **Baseline** | 40.65     | 58.23     | 37.04     | 42.08     | **23.43** | 12.06                | 48.23      | **40.69**  | **39.85** | 56.10        | 48.81     |
> | **+ ours**   | **41.45** | **59.33** | **38.32** | **43.26** | 23.10     | **12.82**            | **49.70**  | 40.01      | 39.26     | **58.50**    | **50.21** |
>
> We can observe significant improvements in detection data for small objects (e.g., traffic_cone and barrier with 2.40 and 1.40 mAP increase respectively), medium-sized objects (e.g., car with a 1.10 mAP increase), and larger objects (e.g., truck and construction_vehicle with 1.28 and 0.76 mAP increase respectively).
>
> It is worth noting that although the mAP increases for most object categories, there is a slight decline in the performance of motorcycles, bicycles, and trailers. Upon further analysis, we attribute this decline to the poor reconstruction performance of point cloud-based methods for objects with holes, such as the wheels of bicycles. Additionally, we identify challenges in the reconstruction of trailers, given their large and complex structures with diverse goods carried. These lead to insufficient quality in the generated data, causing a slight decline in performance data. In subsequent work, we will address and follow up on resolving these related issues.
>
>
> Finally, we express sincere appreciation for your constructive suggestions, which have contributed to the comprehensiveness of our work.

---

> > ### Comment · Reviewer_ZS78 · 2023-11-22
> >
> > Thanks for the answers and clarification in the rebuttal, which covered most of my concerns. Considering the rebuttal and other reviews, I lean to accept it as the raised concerns can be addressed in the final revision.

---

> > > ### Author Response · Authors · 2023-11-23
> > >
> > > We are glad that our responses solve your concerns. Thank you again for your valuable feedback and suggestions!

---

### Author Response · Authors · 2023-11-21

Dear reviewers:

We are grateful for the insightful and constructive feedback provided by the reviewers. We are pleased to read that our research topic is ***meaningful*** and ***interesting*** [ZS78, m1gL], our method is ***simple*** and ***straightforward*** [v2XH], and our evaluation experiments are ***effective***[m1gL].

Furthermore, we have tried our best to address all the mentioned concerns and problems, making our research more robust and accessible. All the modifications will be incorporated into our manuscript.

Since we are in the last two days of the discussion phase, we are eagerly looking forward to your responses. Please let us know if there are any additional clarifications or experiments that we can offer. We would love to discuss more if any concern still remains.

Best,

Authors

---

### Meta-Review · Area_Chair_pVQT · 2023-12-06

**Metareview:**

The paper receives mixed ratings: 1 borderline reject and 2 borderline acceptance. The main concerns from the reviewers are: 1) experimental results (e.g., detailed performance, improved gains); 2) some technical clarity; 3) complexity and cost of the proposed augmentation way. After rebuttal, some concerns are resolved. The AC took a close look at the paper and rebuttal, and finds the limitation of the proposed augmentation to be applicable to wider use cases. For example, the method requires certain specific labels, e.g., driving trajectory of the ego vehicle and the bounding boxes of objects, in which such data may not be available at the same time (e.g., on KITTI). Therefore, it is a bit difficult to understand the practical contribution of the proposed method, given that experiments are only conducted on nuScenes with marginal performance gains. Hence the AC would recommend the rejection rating.

**Justification For Why Not Higher Score:**

There is a specific requirement of the data that can be applied to the proposed augmentation method, which would heavily limit the practical usage.

**Justification For Why Not Lower Score:**

N/A

---

### Decision · Program_Chairs · 2024-01-16

Reject